# ON BITS AND BANDITS: QUANTIFYING THE REGRET-INFORMATION TRADE-OFF

**Itai Shufaro**
Technion
itai.shufaro@campus.technion.ac.il

**Nadav Merlis**
ENSAE
nadav.merlis@ensae.fr

**Nir Weinberger**
Technion
nirwein@technion.ac.il

**Shie Mannor**
Technion, NVIDIA Research
shie@ee.technion.ac.il

## ABSTRACT

In many sequential decision problems, an agent performs a repeated task. He then suffers regret and obtains information that he may use in the following rounds. However, sometimes the agent may also obtain information and avoid suffering regret by querying external sources. We study the trade-off between the information an agent accumulates and the regret it suffers. We invoke information-theoretic methods for obtaining regret lower bounds, that also allow us to easily re-derive several known lower bounds. We introduce the first Bayesian regret lower bounds that depend on the information an agent accumulates. We also prove regret upper bounds using the amount of information the agent accumulates. These bounds show that information measured in *bits*, can be traded off for regret, measured in reward. Finally, we demonstrate the utility of these bounds in improving the performance of a question-answering task with large language models, allowing us to obtain valuable insights.

## 1 INTRODUCTION

In interactive decision-making problems, an agent repeatedly interacts with a task by sequentially choosing decisions from a decision space. Subsequently, the agent receives feedback that usually includes a reward and, optionally, other types of information. For example, the feedback includes only the reward in multi-armed bandit (MAB) problems (Lattimore & Szepesvári, 2020). In partial monitoring, however, it only includes a signal, which the agent then utilizes to indirectly infer the reward (Cesa-Bianchi et al., 2006). The goal of the agent in such tasks is to minimize the gap between the accumulated reward and the reward of the best decision in hindsight, also known as regret.

In these tasks, agents can accumulate information from multiple sources, that can be categorized into two main types. One is information acquired through direct interactions and receiving feedback from the task. For example, in reinforcement learning (RL) such information includes visited states, accumulated rewards, and performed actions. The other type is information provided by external sources, such as human advice (Najar & Chetouani, 2021), text description generated by large language models (Du et al., 2023; Hu & Sadigh, 2023), and more.

In some tasks both information from direct interactions and external information are present. One example is RL with LLM (Du et al., 2023), where an agent interacts with an online environment and is provided with advice from an LLM. The agent can interact with the environment as well as ask the LLM for advice.

Consider two agents playing on the same online task. Both agents have access to the same observations, but the first agent has already played multiple rounds in the task. Thus, he already suffered regret and gained an advantage over the second agent. Before he starts interacting with the task, the second agent can use external knowledge that will allow him to reach the same performance level as the first agent. In this paper, we focus on how much information the second agent needs to query

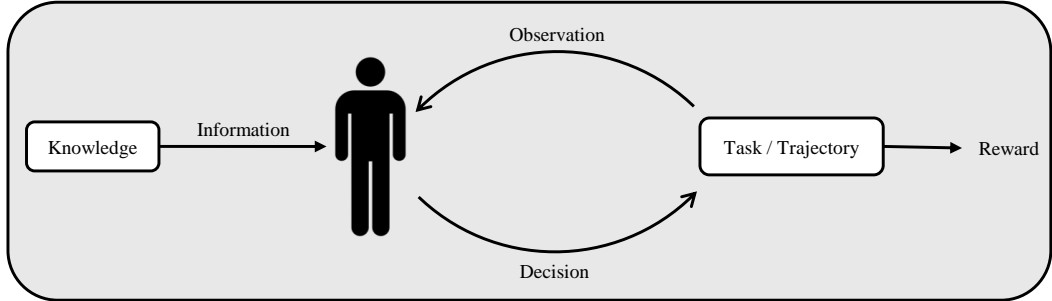

Figure 1: Schematic of a general interactive decision-making task with contextual information. Every round, a source of knowledge provides our agent with information. The agent then makes a decision, that causes the task to generate an observation and a reward. The observation is revealed to the agent, who updates his next decision according to the past observations and the information he received.

so he will "catch up" with the first agent without suffering regret. In other words, we focus on the following question:

*What is the exact relationship between information bits and regret?*

## 1.1 CONTRIBUTIONS

**New method for obtaining regret lower bounds.** We consider an interactive decision-making framework that describes general online tasks with contextual information in a Bayesian setting. We then introduce a novel information-theoretic method to obtain regret lower bounds for decision-making problems in this setting. This method uses Fano's inequality (Cover, 1999) and the construction method introduced by Yang & Barron (1999) to lower bound the worst-case Bayesian regret. In Theorem 3.4 we introduce the general lower bound, and Table 1 shows its application on MAB, contextual bandit, and RL tasks.

**Information-theoretic prior-dependent bounds.** The total amount of information an agent gathers can be quantified to bits by the mutual information functional. Proposition 4.1 presents Bayesian regret lower bounds that depend on the amount of information an agent accumulates. Additionally, Proposition 4.3 upper bounds the regret for Thompson sampling (Thompson, 1933) in an MAB environment, which depends on the information the agent collects. We also present lower bounds for scenarios where the entropy of the prior is constrained in Proposition 4.5.

**Regret-information trade-off.** We show that the presented lower bounds quantify the relationship between external information and regret. Furthermore, these bounds quantify the relationship between the rate of information accumulation and regret in online tasks. These relationships allow us to measure how much regret can be avoided by looking at the information that can be accumulated. We then demonstrate how to easily utilize this insight in an online question-answering task with large language models.

The rest of the paper is organized as follows: In Section 2 we present our setting and relevant preliminaries. In Section 3 we present a novel information-theoretic method for obtaining regret lower bounds. In Section 4 we present regret bounds that depend on the information accumulated by the agent, followed by experiments in Section 5. In Section 6 we review related work, and draw conclusions in Section 7.

## 2 SETTING AND PRELIMINARIES

We introduce a new setting called interactive decision-making in Bayesian environments with contextual information, which is illustrated in Figure 1. This setting is a Bayesian adaptation of the frequentist interactive decision-making setting introduced by Foster et al. (2021). We denote the set

of Borel probability measures over a locally compact space by $\Delta(\cdot)$. Let $\Pi$ be a compact decision space, $\mathcal{M}$ be a compact model space and $\mathcal{C}$ be a compact context space. Every model $M \in \mathcal{M}$ is a mapping from the decision and context space to an observation space $\mathcal{O}$, i.e., $M : \Pi \times \mathcal{C} \to \mathcal{O}$. Every model $M$ has a reward function, $R_M : \Pi \times \mathcal{C} \to \Delta([0,1])$ that maps every decision-context pair to a reward distribution over $[0,1]$. A task in this setting is defined by the decision space $\Pi$, the model class $\mathcal{M}$, the context space $\mathcal{C}$, the reward functions $R_M$, and the prior $P \in \Delta(\mathcal{M} \times \mathcal{C})$. We denote by $\mu_M(\pi, C) = \mathbb{E}_{x \sim R_M(\pi,C)}[x]$ the mean reward of decision $\pi$ under model $M$ and context $C$. With a slight abuse of notation, we also denote by $\mu_M(\phi, C) = \mathbb{E}_{\pi \sim \phi}[\mu_M(\pi, C)]$, the mean reward of a stochastic decision $\phi \in \Delta(\Pi)$ under model $M$ and context $C$.

In a $T$-round game, a random model $M$ is sampled before the game starts according to the prior $P$. Then a context $C_t$ is sampled every round, independently from previous rounds, according to the marginal prior probability $P(\cdot \mid M)$. Before each round starts, the context $C_t$ is revealed to the agent, who determines his stochastic decision $p_t : \mathcal{C} \to \Delta(\Pi)$ according to the context and the observed history. Then, a decision $\pi_t \sim p_t$ is sampled and an observation $O_t \sim M(\pi_t, C_t)$ is revealed to the agent. A reward, $r_t \sim R_M(\pi_t, C_t)$ is also generated. We denote the history up to time $t$ as $\mathcal{H}_t = \{C_i, \pi_i, O_i\}_{i=1}^t$. We denote $\pi_{M,C}^* = \arg\max_{\pi \in \Pi} \mu_M(\pi, C)$. We measure the performance with respect to the best context-dependent decision in hindsight. This performance metric is called the Bayesian regret and is defined by

$$\mathcal{BR}_P(T; \{p_t\}_{t=1}^T) = \mathbb{E}\left[\sum_{t=1}^T \left(\mu_M(\pi_{M,C_t}^*, C_t) - r_t\right)\right] \tag{1}$$

where the expectation is taken with respect to the randomness in decisions, contexts, observations, and rewards $((M, C_t) \sim P, \pi_t \sim p_t$ and $r_t \sim R_M(\pi_t, C_t))$. We also note that by fixing a single context, $|\mathcal{C}| = 1$, the setting reduces to regular Bayesian interactive decision-making. When obvious from the context, $P$ and $\{p_t\}_{t=1}^T$ are omitted. We next show how our general framework covers common decision-making tasks with contextual information.

**Example 2.1** (Contextual MAB with Bernoulli rewards). The contextual MAB problem is defined by the rewards of each arm, which are, say, Bernoulli random variables. For simplicity, we assume a finite set of decisions, which are called arms in this setting, $\Pi = \{1, 2, \ldots, K\}$. Furthermore, we assume that the context space is finite as well, $\mathcal{C} = \{1, 2, \ldots, C\}$. The model class is then given by $\mathcal{M} = \{f | f : \Pi \times \mathcal{C} \to [0,1]\}$, so every model maps $(\pi, C)$ to $\mathrm{Bern}(f(\pi, C))$. At each round $t$, a context is sampled and revealed. A decision $\pi_t$ is selected and the observation is the incurred reward, $r_t = O_t \sim \mathrm{Bern}(f_t(\pi_t, C_t))$. We note that the Bernoulli rewards were selected to maintain a simple example, and that our setting also covers general reward distributions.

**Example 2.2** (Tabular reinforcement learning with a finite horizon). The finite-horizon tabular MDP is defined by the tuple $(\mathcal{S}, \mathcal{A}, \mathbb{P}, \mathbb{R}, H)$ (which are the state space, action space, transition kernel, reward function and horizon, accordingly). We assume the episodic and stationary setting, where we update the policy only at the beginning of every episode. Additionally, we assume that the reward of every episode is bounded in $[0,1]$. In this setting, the decision space is all of the deterministic policies, so $\Pi = \{\pi : \mathcal{S} \times \mathcal{H} \to \mathcal{A}\}$. The model class $\mathcal{M}$ is defined by the set of all MDPs with the same state space, action set, and horizon. At every time step, the agent determines a policy and plays it over the entire horizon. The observations are the trajectories, which include the visited states, rewards, and actions performed.

**Covering and packing.** Consider a normed space $A$ with metric $d$ induced by the norm, and let $\epsilon > 0$. We denote by $\mathcal{N}(d, A, \epsilon)$ and $\mathcal{M}(d, A, \epsilon)$ the $\epsilon$-covering and $\epsilon$-packing numbers of $A$ under the norm $d$, respectively. We define an $\epsilon$-ball of $b \in A$ as $B(b, \epsilon) = \{a \in A : d(b, a) \leq \epsilon\}$, for every $b \in A$ and $\epsilon > 0$. The local packing number, denoted by $\mathcal{M}^{\mathrm{loc}}(d, A, \epsilon)$, is the largest $(\epsilon/2)$-packing set of any set $B(b, \epsilon)$, i.e.,

$$\mathcal{M}^{\mathrm{loc}}(d, A, \epsilon) = \max\{M : \text{there is } b \text{ such that the } (\epsilon/2)\text{-packing number of } B(b, \epsilon) \text{ is } M\}.$$

When $d(x, y) = \|x - y\|_p$ we denote by $\mathcal{M}_p(A, \epsilon), \mathcal{N}_p(A, \epsilon)$ and $\mathcal{M}_p^{\mathrm{loc}}(A, \epsilon)$ the $\epsilon$-packing, $\epsilon$-covering and $\epsilon$-local packing number, respectively. Detailed definitions and properties of the covering, packing and local packing numbers are provided in Appendix B.

**Additional notations.** We denote by $H(X) = \mathbb{E}[-\log Q(X)]$ the Shannon entropy of a random variable $X \sim Q$, and with a slight abuse of notation, we also denote $H(Q) = H(X)$. Given

Table 1: Lower bounds for online problems, recovered using methods presented in the paper.

| Problem | Retrieved lower bound | Optimal? |
|---|---|---|
| MAB | $\Omega(\sqrt{KT})$ | Yes (Audibert & Bubeck, 2009) |
| Tabular RL | $\Omega(\sqrt{HSKT})$ | Yes (Zhang et al., 2021) |
| Linear bandits | $\Omega(\sqrt{dT})$ | No |
| Lipschitz bandit | $\Omega\left(T^{\frac{d+1}{d+2}}\right)$ | Yes (Kleinberg et al., 2019) |
| **Bayesian MAB ($R$ bits)** | $\Omega\left(\sqrt{KT\frac{\log K}{R}}\right)$ | First bound |
| **Bayesian linear MAB ($R$ bits)** | $\Omega\left(\sqrt{dT\frac{\log K}{R}}\right)$ | First bound |
| **Bayesian MAB** | $\Omega\left(\sqrt{KT\frac{H(\pi^*)}{\log K}}\right)$ | First bound |

two jointly distributed random variables, $(X, Y) \sim Q$, we denote their marginal distributions by $Q(x)$ and $Q(y)$. We also define the mutual information of two jointly distributed random variables $(X, Y) \sim Q$ as $I(X; Y) = \mathbb{E}\left[\log\left(\frac{Q(X,Y)}{Q(X)Q(Y)}\right)\right]$. The KL-divergence is defined as $D_{\mathrm{KL}}\left(P \parallel Q\right) = \mathbb{E}_{X \sim P}\left[\log\frac{P(X)}{Q(X)}\right]$ (Cover, 1999). Unless stated otherwise, logarithms are assumed to be in base-2, denoted by log. We also denote $x \gtrsim y$ if there is some global constant $c > 0$ such that $x \geq cy$. We define by $\mathbb{B}_p^d = \{x \in \mathbb{R}^d : \|x\|_p \leq 1\}$ the unit sphere in $\mathbb{R}^d$ under the $L_p$ norm.

## 3 REGRET LOWER BOUNDS USING FANO'S INEQUALITY

We now introduce a novel approach for obtaining worst-case Bayesian regret lower bounds, which can be applied to a wide variety of interactive decision-making tasks. This method can be viewed as an extension of Fano's inequality for non-exact recovery (Scarlett & Cevher, 2019) to online problems. We define worst-case Bayesian regret in the following manner,

$$\sup_{\nu \in \Delta(\mathcal{M} \times \mathcal{C})} \inf_{\{p_t\}_{t=1}^T} \mathcal{BR}_\nu(T; \{p_t\}_{t=1}^T) \equiv \mathcal{BR}^*(T). \tag{2}$$

To lower bound the worst-case Bayesian regret we use Fano's inequality.

**Theorem 3.1** (Fano's inequality, Theorem 2.10 of Cover (1999)). *Let $X, Y \sim Q$ be two jointly distributed random variables, where $X$ can take values over a finite set, whose cardinality is $\mathcal{X}$. Let $\hat{X} = f(Y)$ for some $f$ be an estimator of $X$. If $\hat{X}$ is uniformly distributed over all possible values in $\mathcal{X}$, then the following holds for all $f$,*

$$\mathbb{P}(X \neq \hat{X}) \geq 1 - \frac{I(X; Y) + 1}{\log \mathcal{X}}. \tag{3}$$

While Theorem 3.1 is used to lower-bound the error in hypothesis testing problems, we now show how it can also be applied to regret minimization. We assume access to some set $\Phi = \{\phi_1, \ldots, \phi_K\}$ of (possibly stochastic) decisions. We also assume that a metric $\rho$ exists such that $\mathbb{E}[|\mu_M(\psi, C) - \mu_M(\phi, C)|] \geq \rho(\phi, \psi)$ for all $\psi, \phi \in \Delta(\Pi)$. The following proposition applies Theorem 3.1 to regret minimization, by reducing it to a hypothesis testing over the finite set $\Phi$.

**Proposition 3.2.** *For any algorithm, prior $P$, a decision set $\Phi = \{\phi_1, \ldots, \phi_K\} \subseteq \Delta(\Pi)$ and for all $\epsilon > 0$ such that $\rho(\phi_i, \phi_j) \geq \epsilon$ for all $i \neq j$,*

$$\frac{\mathcal{BR}_P(T)}{T\epsilon} \geq \frac{1}{2}\left[1 - \frac{I(V; \mathcal{H}_T) + 1}{\log K}\right] \tag{4}$$

*where $V$ is the index of the best decision in hindsight from the set $\Phi$.*

The proof of Proposition 3.2 is provided in Appendix C.1. Proposition 3.2 allows us to lower-bound the Bayesian regret using a specific set of decisions. We suggest a method for selecting the

decision set $\Phi$, which provides a general approach for obtaining regret lower bounds. We separate our analysis into finite, parametric with infinite decisions, and non-parametric decision spaces. This separation is done to make the lower bound in Equation 4 tighter. We observe that the lower bound is tighter for larger values of $K$. Thus, we should select the largest value of $K$ that satisfies the requirement $\rho(\phi_i, \phi_j) \geq \epsilon$ for a given $\epsilon > 0$. For finite decision spaces, we choose $\Phi$ to be an $\epsilon$-local packing set of the decision simplex, $\Delta(\Pi)$, according to $\rho$, under $P$. So, we have $K = \mathcal{M}^{\mathrm{loc}}(\rho, \Delta(\Pi), \epsilon)$. Similarly, for parametric decision spaces with infinite decisions, we choose $\Phi$ to be an $\epsilon$-local packing set of the decision set, $\Pi$, i.e., $K = \mathcal{M}^{\mathrm{loc}}(\rho, \Pi, \epsilon)$. For non-parametric decision spaces, we use an $\epsilon$-global packing set of our decision space $\Pi$, which means that $K = \mathcal{M}(\rho, \Pi, \epsilon)$. After selecting $\Phi$, we need to handle $I(V; \mathcal{H}_T)$. To do this, we make the following assumption about the decisions.

**Assumption 3.1.** *There exist constants $\bar{A}, \epsilon_0 > 0$ such that for all stochastic decisions $p_1, p_2$ such that $\rho(p_1, p_2) \leq \epsilon_0$, $D_{\mathrm{KL}}(p_1 \parallel p_2) \leq 2\bar{A}\rho(p_1, p_2)^2$.*

This assumption means that for all stochastic decisions in an $\epsilon_0$-ball around $p_1$, the KL divergence can be upper bounded using $\rho$, and is commonly made, for example, by Yang & Barron (1999). The following theorem upper bounds $I(V; \mathcal{H}_T)$.

**Theorem 3.3** (Yang & Barron (1999)). *Let Assumption 3.1 hold. Under the assumptions and notations of Proposition 3.2, if $\Pi$ is a parametric decision space then*

$$I(V; \mathcal{H}_T) \leq 2\bar{A}\epsilon^2 T \tag{5}$$

*for $\epsilon \leq \epsilon_0$. If $\Pi$ is a non-parametric decision space then*

$$I(V; \mathcal{H}_T) \leq \inf_{\delta > 0} \left( \log \mathcal{N}\left(\rho, \Pi, \sqrt{\delta}\right) + T\delta \right). \tag{6}$$

Now, substituting Theorem 3.3 in Proposition 3.2, and selecting $\Phi$ as we described above, results in the following Theorem.

**Theorem 3.4.** *Let there be a Bayesian interactive decision-making problem as defined in Section 2, and let $\epsilon > 0$. If $\Pi$ is a finite decision space,*

$$\frac{\mathcal{BR}^*(T)}{T\epsilon} \geq \frac{1}{2}\left[ 1 - \frac{2\epsilon^2 \bar{A} T + 1}{\log \mathcal{M}^{\mathrm{loc}}(\rho, \Delta(\Pi), \epsilon)} \right]. \tag{7}$$

*If $\Pi$ is an infinite parametric decision space,*

$$\frac{\mathcal{BR}^*(T)}{T\epsilon} \geq \frac{1}{2}\left[ 1 - \frac{2\epsilon^2 \bar{A} T + 1}{\log \mathcal{M}^{\mathrm{loc}}(\rho, \Pi, \epsilon)} \right]. \tag{8}$$

*If $\Pi$ is a non-parametric decision space,*

$$\frac{\mathcal{BR}^*(T)}{T\epsilon} \geq \frac{1}{2}\left[ 1 - \frac{\inf_{\delta > 0}\left( \log \mathcal{N}\left(\rho, \Pi, \sqrt{\delta}\right) + T\delta \right) + 1}{\log \mathcal{M}(\rho, \Pi, \epsilon)} \right]. \tag{9}$$

The proof of Theorem 3.4 is provided in Appendix C.2. To obtain regret lower bounds, all we need are the values of the packing or local packing, and covering numbers of the decision space. Then, we simply need to select a value of $\epsilon$ such that the right-hand side of Equations 7, 8 or 9 is a positive constant. In Appendix C.3 we demonstrate this, by deriving regret lower bounds for several known online problems, which are summarized in Table 1. Moreover, this method can be used to obtain lower bounds for the frequentist regret, due to the mini-max theorem, which states that the worst-case Bayesian regret is equal to the mini-max frequentist regret (Lattimore & Szepesvári, 2019).

## 4 INFORMATION THEORETIC BAYESIAN REGRET UPPER AND LOWER BOUNDS

### 4.1 MUTUAL INFORMATION CONSTRAINT

We now derive Bayesian regret bounds under a constraint on the total amount of information that the agent accumulates. Intuitively, in information-theoretic terms, mutual information quantifies

the amount of information one random variable contains about another one (Cover, 1999). This means that mutual information can be utilized to measure the information that is aggregated from various sources using a common measure of bits. We are interested in lower bounding the regret, under a constraint on the amount of information the agent accumulates. Then, we ask the following question: What is the lower bound of the worst-case Bayesian regret, when the agent accumulates less than $R$ bits of information? We will use the method described in Section 3 to derive regret lower bounds under this constraint. We note that in general $I(\pi^*; \mathcal{H}_T) = D_{\mathrm{KL}}\left(P^{\pi^*|\mathcal{H}_T} \parallel P^{\pi^*}\right)$. Thus, for general agents, $D_{\mathrm{KL}}\left(p_t \parallel P^{\pi^*}\right)$ is the information accumulated up to round $t$, in bits. The following proposition answers this question, for problems with a finite decision space.

**Proposition 4.1.** *Let $0 < R \le \log K$ be given. Assume that the Bayesian decision problem has a finite decision space of size $K$, and that the agent accumulates no more than $R$ bits. Then the worst-case Bayesian regret can be lower bounded by $\Omega\left(\sqrt{TK\frac{\log K}{R}}\right)$. Additionally, if $\Pi \subset \mathbb{B}_2^d$ and the rewards are a linear function of the decisions, then the worst-case Bayesian regret can be lower bounded by $\Omega\left(\sqrt{Td\frac{\log K}{R}}\right)$. Furthermore, if $R = 0$ the worst-case regret is linear.*

The proof of Proposition 4.1 utilizes Lemma D.1, which, similarly to the proof of Theorem 3.3, introduces an upper bound for the mutual information, but this time taking the information constraint into account. The rest of the proof follows a similar structure to the proof of Theorem 3.4, substituting the tighter bound into Equation 7. This yields the following proposition.

**Proposition 4.2.** *Let $R \le \log K$ be given. Assume that the Bayesian decision problem has a finite decision space of size $K$, such that $D_{\mathrm{KL}}\left(p_t \parallel P^{\pi^*}\right) \le R$ for all $t$. Then,*

$$\frac{\mathcal{BR}^*(T)}{T\epsilon} \ge \frac{1}{2}\left[1 - \frac{2\epsilon^2 \bar{A}T\frac{R}{\log K} + 1}{\log \mathcal{M}^{\mathrm{loc}}(\rho, \Delta(\Pi), \epsilon)}\right] \tag{10}$$

*Furthermore, if the decision-making problem is a part of a parametric set, then*

$$\frac{\mathcal{BR}^*(T)}{T\epsilon} \ge \frac{1}{2}\left[1 - \frac{2\epsilon^2 \bar{A}T\frac{R}{\log K} + 1}{\log \mathcal{M}^{\mathrm{loc}}(\rho, \Pi, \epsilon)}\right] \tag{11}$$

The proof follows from Proposition 4.2, by selecting an appropriate value of $\epsilon$. We now state an upper bound for MAB problems where the agent accumulates $R$ bits.

**Proposition 4.3.** *Assume an $K$-MAB problem with no constraints. Let $\tilde{O}_t, \tilde{\mathcal{H}}_T$ be constrained observations and history such that $I(\pi^*; \tilde{\mathcal{H}}_T) = R$. The regret of Thompson sampling with these observations is upper bounded by $O\left(\log K\sqrt{\frac{KT}{R}}\right)$. Additionally, if the MAB is linear the regret can be upper bounded by $O\left(\log K\sqrt{\frac{dT}{R}}\right)$.*

Detailed proofs of Propositions 4.1, 4.2, and 4.3 can be found in Appendix D.

Now, we can quantify the relationship between information and regret. Consider two agents, $A$ and $B$, that play on the same online decision-making task. $A$ has an advantage over $B$, since he already played multiple rounds before. $B$ can query external information to reach the same level of performance as $A$ on the task. Using the regret upper bound in Proposition 4.3 we can get a lower bound on the information $B$ needs. Similarly, Proposition 4.1 gives an upper bound on the number of bits. Thus, Propositions 4.1 and 4.3 quantify information in terms of regret. We note that this quantification is not tight, as the upper bounds differ from the lower bounds by a factor of $\sqrt{\log K}$.

These Propositions also allow us to quantify the relationship between the rate of information accumulation and regret. If both parties play for the same number of rounds, the one that accumulates information faster will also suffer less regret.

## 4.2 ENTROPY CONSTRAINT

While Proposition 4.1 shows how we can lower bound regret under a constraint on accumulated information, we now shift our focus to a constraint on the prior only, i.e $H(\pi^*) \le R$. Russo

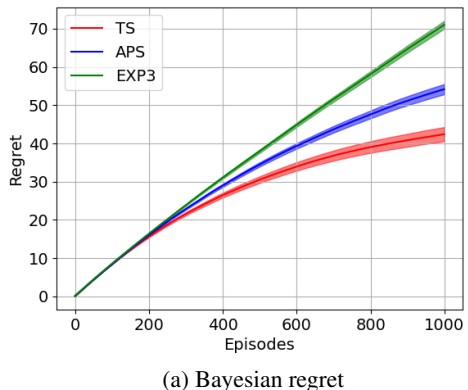
(a) Bayesian regret

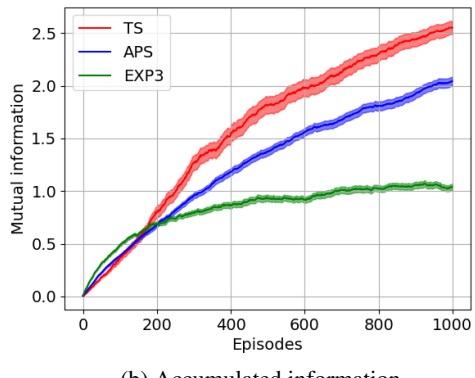
(b) Accumulated information

Figure 2: (a) The Bayesian regret and (b) the accumulated information in bits for three different bandit algorithms, under the same bandit structure and a uniform prior. The bandit algorithms used are described in the legend. The shadowed areas correspond to 2-sigma error bars.

& Van Roy (2014; 2016) showed that in this scenario, the regret of Thompson sampling and information-directed sampling (IDS) can be upper bounded using $H(\pi^*)$.

**Theorem 4.4** (Propositions 2 and 4 of Russo & Van Roy (2014)). *In the $K$-MAB setting, the regret of Thompson sampling and IDS is upper bounded as $O(\sqrt{KTH(\pi^*)})$. Furthermore, if the rewards are a linear function of the decisions, and the decisions are vectors in $\mathbb{R}^d$, then the regret is upper bounded by $O\left(\sqrt{dTH(\pi^*)}\right)$.*

We next develop an entropy-dependent Bayesian regret lower-bound for algorithms that obey the following assumption.

**Assumption 4.1.** $D_{\mathrm{KL}}\left(p_t \parallel P^{\pi^*}\right) \geq c\bar{A}R$ *for all $t$, for some constant $c > 0$.*

Intuitively, this assumption means that we consider only agents that gather a minimal amount of information regarding the problem. We now state the following theorem.

**Proposition 4.5.** *For any agent for the $K$-MAB problem that satisfy Assumption 4.1, the regret is lower-bounded as $\Omega\left(\sqrt{\frac{KH(\pi^*)T}{\log K}}\right)$. Additionally, if the MAB is linear, then the regret is lower bounded by $\Omega\left(\sqrt{\frac{dH(\pi^*)T}{\log K}}\right)$.*

The proof of Proposition 4.5 can be found in Appendix D.3. We note that this lower bound behaves like the upper bound Presented in Theorem 4.4, up to a factor of $\sqrt{\log K}$. We also show in the appendix that Assumption 4.1 holds for any algorithm that learns the optimal decision.

## 5 EXPERIMENTS

### 5.1 STOCHASTIC BAYESIAN BANDIT

We begin by demonstrating the trade-off in a simple Bayesian MAB problem. In this problem, our model space contains $K$ possible models. The decision space of the problem also contains $K$ decisions, which are called arms. The mean of all arms in every model is $0.5(1 - \varepsilon)$, except for one arm for which the mean is $0.5(1 + \varepsilon)$. The optimal arm is different for every model, and the reward of each arm is Bernoulli distributed. We also fix $K = 8, \varepsilon = 0.1$ and a uniform prior over the problems and compare three different bandit algorithms - EXP3 (Lattimore & Szepesvári, 2020), APS (Xu & Zeevi, 2023), and Thompson sampling (Thompson, 1933). We also estimate the accumulated information for each algorithm. Results are presented in Figure 2. We see how algorithms that accumulate information quickly also suffer less regret. We also compare Thompson

Table 2: Mean regret for both policies for a horizon of 200 episodes, under different deployment percentages of the large LLM. The left column denotes the percentage of queries to the large LLM. For all tables, we used Mixtral-8x7B as the large LLM. For the small LLM, we used Mistral-7B (top table) and Gemma-2B (bottom table). Comparison with other LLMs is provided in Appendix F.

|  | % Deployment | Bits-based | Random | Small only | Large only |
|---|---|---|---|---|---|
| Mistral 7B | 50% | **36 ± 1** | 38.7 ± 0.3 | 59.3±0.7 | 18.2±0.6 |
|  | 70% | **26.6 ± 0.8** | 30.5 ± 0.4 |  |  |
|  | 90% | **21.8 ± 0.4** | 22.4 ± 0.5 |  |  |
| Gemma 2B | 50% | **38 ± 2** | 44.5 ± 0.7 | 67 ± 1 | 22 ± 1 |
|  | 70% | **32 ± 1** | 35.6 ± 0.8 |  |  |
|  | 90% | **26 ± 1** | 27 ± 1 |  |  |

sampling under different priors $H(\pi^*)$. The results of this experiment can be found in Figure 3. Additional experimental details are provided in Appendix E.

## 5.2 QUESTION ANSWERING WITH LARGE LANGUAGE MODELS

In the following experiment, we show how the regret-information trade-off can be utilized in a practical setting. We consider a sequential multiple choice question-answering (MCQA) task, where every round we need to answer a question, given four possible answers. We also have access to two large language models (LLM), where one has significantly more parameters than the other. At every turn, we choose which LLM should be used to answer the question. Every round we are provided with the question, possible answers, and the output of the small LLM for the given prompt. We receive a positive reward of 1 for answering the question correctly. If we choose to deploy the large LLM we also incur a negative reward of 0.1. Our goal is to minimize the accumulated regret.

On the one hand, the larger LLM is more accurate and will output the correct answer with a higher probability. On the other hand, we do not wish to query the large LLM due to the penalty we suffer, if the small LLM already outputs the correct answer with high probability. We present the following method for selecting the LLM using bits: Use the small LLM to quantify the amount of information that can be obtained. If it is above some threshold, we opt to use the large LLM since it provides more information, which results in less regret as we have seen in Section 3. Otherwise, use the small LLM. We call this approach the bits-based policy.

Since the LLM outputs a probability distribution over tokens, we can measure the information the small LLM will gain after answering the question, in bits. We prompt the small LLM with the question and possible answers, which returns scores for each token. We take the scores corresponding only for the tokens [A, B, C, D], each corresponding to a different answer, and use them to get a distribution over these tokens. We measure the information in bits by calculating the KL divergence between this distribution and the uniform one. We compare the mean regret over a horizon of 200 steps between this policy, and one that randomly selects which LLM to use. The threshold value for the bits-based policy is selected to ensure that we query the large LLM the same number of times as the random policy.

We run the experiment described above with the following specifications. To generate the multiple-choice questions, we used the MMLU dataset, which contains multiple-choice questions in a variety of topics, such as algebra, business, etc. (Hendrycks et al., 2021b;a). We used 10 different seeds to generate 10 sets of 200 questions from MMLU randomly. For the small LLM we either used Mistral 7B Jiang et al. (2023), Falcon 7B (Almazrouei et al., 2023), or Gemma 2B (Team et al., 2024). We used Mixtral 7Bx8 (Jiang et al., 2024), Llama3-70B (AI@Meta, 2024), or Gemma 7B for the large LLM. We applied 4-bit quantization (Dettmers & Zettlemoyer, 2023) for all models and flash attention (Dao et al., 2022) for all models excluding Falcon 7B. Table 2 describes the mean regret of every policy under a different number of large LLM deployments. Tables 4 and 5 provide additional results for different combinations of small and large LLMs.

From the results, we see that deciding whether to query the large LLM or not using bits is better than random selection. Furthermore, this improvement becomes more significant as we increase the

number of deployments allowed. This demonstrates how we can easily utilize the quantification of information to improve performance in online tasks. Additional details and results for other models are provided in Appendix F. Code is available here[1].

## 6 RELATED WORK

**Bayesian setting.** In the Bayesian setting, prior information is assumed to be expressed as a probability distribution, called the prior. New information is then incorporated using Bayesian inference to update the prior probability, resulting in a posterior distribution. Bayesian algorithms have been extensively studied in multiple online decision-making problems such as multi-armed bandits (Russo & Van Roy, 2014; Kaufmann et al., 2012) and reinforcement learning (Guez et al., 2012; Ghavamzadeh et al., 2016). Upper bounds for Bayesian algorithms have been proved for both the frequentist (Kaufmann et al., 2011) and Bayesian settings (Russo & Van Roy, 2014; 2016). Prior dependent regret bounds for Bayesian algorithms have also been studied (Bubeck & Liu, 2013; Russo & Van Roy, 2014). Previous studies have explored prior-dependent lower bounds for Bayesian regret, but these are limited to specific forms of priors, such as Gaussian priors (Atsidakou et al., 2024).

**Contextual information.** In the contextual setting, information is revealed to the agent before every round, which can be leveraged to minimize regret. This framework has gathered attention within both the bandit (Bietti et al., 2021) and reinforcement learning frameworks (Klink et al., 2020; Modi & Tewari, 2020). The contexts can be selected arbitrarily by an adversary (Beygelzimer et al., 2011; Chu et al., 2011) or generated from some prior probability distribution, similarly to the Bayesian setting (Hao et al., 2020; May et al., 2012). Bayesian algorithms have also been adapted to the arbitrary context setting (Agrawal & Goyal, 2013). Furthermore, the type of information provided to the learner can expand beyond the contextual information provided directly (Schneider & Zimmert, 2024). Our work introduces a contextual Bayesian setting framework that covers a wide variety of interactive decision-making tasks.

**Information-theoretic methods** have been employed to derive upper bounds for various online tasks. Russo & Van Roy (2016) introduced an information-theoretic method for establishing Bayesian regret upper bound for Thompson sampling in the multi-armed bandit setting, which depends on the entropy of the prior. This approach was also used to design new Bayesian algorithms that utilize mutual information called information-directed sampling (IDS) (Russo & Van Roy, 2014). IDS and the information-theoretic method it utilizes were also extended to other tasks such as contextual bandits (Neu et al., 2022), sparse linear bandits (Hao et al., 2021), non-stationary bandits (Min & Russo, 2023; Liu et al., 2023), reinforcement learning (Lu & Van Roy, 2019), non-linear control (Kakade et al., 2020), partial monitoring (Lattimore & Szepesvári, 2019), and other online optimization problems (Liu et al., 2018; Dong et al., 2019; Lattimore & Gyorgy, 2021; Russo & Van Roy, 2018). The bounds presented in works such as (Russo & Van Roy, 2016; 2014; Neu et al., 2022) are based on the upper bounding of the information ratio. Our work uses a different approach, which obtains nearly matching prior-dependent lower bounds for Thompson sampling and information-directed sampling (IDS). Furthermore, these bounds can be applied beyond the bandit setting. Seldin et al. (2011) utilized PAC-Bayes bounds to obtain information-theoretic upper bounds on the per-round regret, that depend on mutual information. Arumugam & Van Roy (2021; 2022) have explored a different information-theoretic method, utilizing rate-distortion theory to minimize regret.

## 7 CONCLUSIONS AND FUTURE WORK

We introduce a general setting that embeds contextual information in a Bayesian setting. This setting covers a wide variety of online decision-making tasks. Using information-theoretic tools, we demonstrated a general method for obtaining regret lower bounds for problems in this setting. We used this method to present regret lower bound which depends on the information accumulated by the agent. We also presented regret upper bounds for Thompson sampling which depends on the

---

[1]https://github.com/itaishufaro/bitsandbandits

accumulated information. These results quantify the relationship between a-priori external information and regret in online settings and the relationship between online information accumulation and regret. We then utilized this trade-off in a multiple-choice question-answering task with LLMs, demonstrating that the aforementioned quantification can be easily used in an online setting.

**Limitations.** A central assumption in our analysis is that exogenous information can be gathered regardless of the process history. However, this assumption may be violated in a general sequential decision process where the action itself may have ramifications on the quality, quantity, and cost of the exogenous information. Our work only covers the Bayesian setting, and our definition of the measure of information relies on it. In other settings, such as adversarial learning (Neu & Olkhovskaya, 2020) or frequentist settings, a different measure of information is required. Another limitation is the difference in $\sqrt{\log K}$ between the lower and upper bounds presented in this work. Making these bounds tighter can be the topic of future work.

Finally, and importantly, with the increasing prevalence of LLMs, and foundation models in general, building solid foundations as well as practical algorithms for using prior knowledge in sequential decision-making is an important research endeavor that may be built upon the foundations that are laid in this paper.

## ACKNOWLEDGEMENTS

This project has received funding from the European Union's Horizon 2020 research and innovation programme under the Marie Skłodowska-Curie grant agreement No 101034255.

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

## A  TABLE OF NOTATIONS

Table 3: Commonly used notations throughout the paper and the appendix.

| Notations | Description |
|---|---|
| $\Pi$ | The decision space. |
| $\mathcal{M}$ | The model class. |
| $\mathcal{O}$ | The observation space. |
| $\mathcal{C}$ | The context space. |
| $\pi$ | A decision. |
| $R_M(\pi, C)$ | reward distribution according to decision $\pi$ on model $M$. |
| $\mu_M(\pi, C)$ | mean reward according to decision $\pi$ on model $M$. |
| $P$ | Prior probability of the task. |
| $\pi_M^*$ | The optimal decision for a model $M$. |
| $\mathcal{H}_t$ | The total accumulated history up to round $t$ (including observations and contexts). |
| $p_t$ | stochastic decision selected at round $t$. |
| $r_t$ | Reward sampled at time $t$. |
| $\rho$ | A norm that lower bounds the difference in reward means. |
| $\mathcal{BR}_P(T)$ | The Bayesian regret given prior $P$. |
| $\mathcal{BR}^*(T)$ | The worst-case Bayesian regret. |
| $\mathcal{N}(d, \epsilon, H)$ | $\epsilon$-covering number of $H$ concerning metric $d$. |
| $\mathcal{M}(d, \epsilon, H)$ | $\epsilon$-packing number of $H$ concerning metric $d$. |
| $\mathcal{M}^{\mathrm{loc}}(d, \epsilon, H)$ | $\epsilon$-local packing number of $H$ concerning metric $d$. |
| $\mathrm{Vol}(\cdot)$ | Volume of a set. |

## B  USEFUL PROPERTIES OF COVERING NUMBERS

**Definition B.1** (Covering and packing numbers (Wainwright, 2019))**.** Let $H$ be a normed space, and let $d(\cdot, \cdot)$ be the metric induced by the norm. Let $\epsilon > 0$.

- A set $C \subset H$ is said to be an $\epsilon$-covering set of $H$ if for all $h \in H$ there is $c \in C$ such that $d(h, c) \leq \epsilon$.

- The covering number is defined as
$$\mathcal{M}(d, \epsilon, H) = \min \{m : \exists C, |C| = m \text{ and } C \text{ is an } \epsilon\text{-covering set of } H\}.$$
If $d(x, y) = \|x - y\|_p$, we denote it by $\mathcal{M}_p(\epsilon, H)$.

- A set $C \subset H$ is said to be a packing set of $H$ if for all $c_1, c_2 \in C$ we have that $d(c_1, c_2) \geq \epsilon$.

- The packing number is defined as
$$\mathcal{N}(d, \epsilon, H) = \max \{m : \exists C, |C| = m \text{ and } C \text{ is an } \epsilon\text{-packing set of } H\}.$$
If $d(x, y) = \|x - y\|_p$, we denote it by $\mathcal{N}_p(\epsilon, H)$.

- We now define $B(\theta, \epsilon) = \{\theta' \in H : d(\theta', \theta) \leq \varepsilon\}$. $C \subseteq B$ is an $\epsilon$-local packing set of $B(\theta, \epsilon)$ if for all $c_1, c_2 \in C$, $d(c_1, c_2) \geq \frac{\epsilon}{2}$.

- The local packing number is defined as

$$\mathcal{M}^{\mathrm{loc}}(d, \epsilon, H) = \max \left\{ m : \exists B(\theta, \varepsilon), \text{such that } C \text{ is an } \epsilon - \text{local packing of } B \text{ and } |C| = m \right\}.$$

**Lemma B.1.** *The following properties hold:*

- *(Theorem 14.2 of Wu (2020)) For any subset of dimension $d$ that is contained in $\mathbb{B}_p^d$ the covering number $\mathcal{N}_p(\epsilon)$ obeys $d \log \left( \frac{2}{\epsilon} \right) \leq \log \mathcal{N}_p(\epsilon) \leq d \log \left( \frac{3}{\epsilon} \right)$. This includes the probability simplex with $d$ variables and $\mathbb{B}_d^2$.*

- *(Theorem 14.1 of Wu (2020)) Under the same norms, for the same set,*

$$\mathcal{M}(2\epsilon) \leq \mathcal{N}(\epsilon) \leq \mathcal{M}(\epsilon)$$

Throughout the appendix, $\mathrm{Vol}(A)$ denotes the volume of set $A$.

## C  PROOFS FOR SECTION 3

### C.1  PROOF OF PROPOSITION 3.2

*Proof.* We write

$$
\begin{aligned}
\mathcal{BR}^*(T) &= \sup_{\nu} \inf_{p_1,\ldots,p_T} \mathbb{E}_{M \sim \nu} \left[ \sum_{t=1}^{T} (\mu_M(\pi^*, C_t) - \mu_M(p_t, C_t)) \right] \\
&\geq \sup_{\nu} \inf_{p_1,\ldots,p_T} \frac{\epsilon T}{2} \mathbb{P}_\nu \left( \sum_{t=1}^{T} (\mu_M(\pi^*, C_t) - \mu_M(p_t, C_t)) \geq \frac{\epsilon T}{2} \right) \\
&\geq \frac{\epsilon T}{2} \sup_{\nu} \inf_{p_1,\ldots,p_T} \mathbb{P}_\nu \left( \forall t : \mu_M(\pi^*, C_t) - \mu_M(p_t, C_t) \geq \frac{\epsilon}{2} \right) \\
&\geq \frac{\epsilon T}{2} \max_{k \in \{1,\ldots,K\}} \inf_{p_1,\ldots,p_T} \mathbb{P} \left( \forall t : |\mu_M(\phi_k, C_t) - \mu_M(p_t, C_t)| \geq \frac{\epsilon}{2} \right) \\
&\geq \frac{\epsilon T}{2} \max_{k \in \{1,\ldots,K\}} \inf_{p_1,\ldots,p_T} \mathbb{P} \left( \forall t : \rho(\phi_k, \pi_t) \geq \frac{\epsilon}{2} \right) \\
&\geq \frac{\epsilon T}{2} \max_{k} \inf_{t} \mathbb{P}(p_t \neq \phi_k) \\
&\geq \frac{\epsilon T}{2} \left( 1 - \frac{I(V; \mathcal{H}_T) + 1}{\log K} \right).
\end{aligned}
$$

The first inequality follows from Markov's inequality, the second inequality from a union bound, and the last inequality from Fano's inequality (Theorem 3.1). □

### C.2  PROOF OF THEOREM 3.4

*Proof.* We provide proof for three different scenarios.

- First scenario, finite $\Pi$. For this scenario, we choose $\Phi$ to be a local packing set of the decisions simplex $\Delta(\Pi)$, so $K = \mathcal{M}^{\mathrm{loc}}(\rho, \Delta(\Pi), \epsilon)$. Applying this selection of set, and using Equation 4 we have that

$$\frac{\mathcal{BR}^*(T)}{\epsilon T} \geq \frac{1}{2} \left[ 1 - \frac{1 + I(V; \mathcal{H}_T)}{\log \mathcal{M}^{\mathrm{loc}}(\rho, \Delta(\Pi), \epsilon)} \right]. \tag{12}$$

Applying Theorem 3.3 we have that $I(V; \mathcal{H}_T) \leq 2\bar{A}\epsilon^2 T$. Substituting this back into Equation 12,

$$\frac{\mathcal{BR}^*(T)}{\epsilon T} \geq \frac{1}{2} \left[ 1 - \frac{1 + 2\bar{A}\epsilon^2 T}{\log \mathcal{M}^{\mathrm{loc}}(\rho, \Delta(\Pi), \epsilon)} \right].$$

- Second scenario, infinite-parametric $\Pi$. For this scenario, we choose $\Phi$ to be a local packing set of the decision space $\Pi$, so $K = \mathcal{M}^{\mathrm{loc}}(\rho, \Pi, \epsilon)$. Applying this selection of set, and using Equation 4 we have that

$$\frac{\mathcal{BR}^*(T)}{\epsilon T} \geq \frac{1}{2} \left[ 1 - \frac{1 + I(V; \mathcal{H}_T)}{\log \mathcal{M}^{\mathrm{loc}}(\rho, \Pi, \epsilon)} \right]. \tag{13}$$

Applying Theorem 3.3 we have that $I(V; \mathcal{H}_T) \leq 2\bar{A}\epsilon^2 T$. Substituting this back into Equation 13,

$$\frac{\mathcal{BR}^*(T)}{\epsilon T} \geq \frac{1}{2} \left[ 1 - \frac{1 + 2\bar{A}\epsilon^2 T}{\log \mathcal{M}^{\mathrm{loc}}(\rho, \Pi, \epsilon)} \right].$$

- Third scenario, non-parametric $\Pi$. For this scenario, we choose $\Phi$ to be a global packing set of the decision space $\Pi$, so $K = \mathcal{M}(\rho, \Pi, \epsilon)$. Applying this selection of set, and using Equation 4 we have that

$$\frac{\mathcal{BR}^*(T)}{\epsilon T} \geq \frac{1}{2} \left[ 1 - \frac{1 + I(V; \mathcal{H}_T)}{\log \mathcal{M}(\rho, \Pi, \epsilon)} \right]. \tag{14}$$

Applying Theorem 3.3 we have that $I(V; \mathcal{H}_T) \leq \inf_{\delta > 0} \left( \log \mathcal{N}\left(\rho, \Pi, \sqrt{\delta}\right) + T\delta \right)$. Substituting this back into Equation 14,

$$\frac{\mathcal{BR}^*(T)}{\epsilon T} \geq \frac{1}{2} \left[ 1 - \frac{1 + \inf_{\delta > 0} \left( \log \mathcal{N}\left(\rho, \Pi, \sqrt{\delta}\right) + T\delta \right)}{\log \mathcal{M}(\rho, \Pi, \epsilon)} \right].$$

$\square$

## C.3 Proofs for Additional Lower Bounds

**Proposition C.1.** *For any algorithm, there exists a Bayesian decision-making task with a finite decision set and a single context ($|\mathcal{C}| = 1$) such that the regret can be lower bounded by $\Omega(\sqrt{|\Pi| T})$.*

*Proof.* We focus on the set $\Delta(\Pi)$ and note that it is a parametric set. We also see that for all stochastic decisions $\phi, \psi \in \Delta(\Pi)$ we have that

$$\mathbb{E}[|\mu_M(\phi, C) - \mu_M(\psi, C)|] = c \|\phi - \psi\|_1.$$

for some positive constant $c$. Hence, in this scenario, we can take $\rho(x, y) = c \|x - y\|_1$. It is convenient to denote $K = |\Pi|$. Thus, using a scaling argument,

$$\log \mathcal{M}^{\mathrm{loc}}(\rho, \Delta(\Pi), \epsilon) = \log \mathcal{M}_1^{\mathrm{loc}}(\Delta(\Pi), \epsilon/c).$$

From the definition of a local packing number, we have that

$$\log \mathcal{M}_1^{\mathrm{loc}}(\Delta(\Pi), \epsilon/c) \geq \log \left( \frac{\mathrm{Vol}(\mathbb{B}_1^{K-1}(\epsilon))}{\mathrm{Vol}(\mathbb{B}_1^{K-1}(\epsilon/2))} \right)$$

$$\geq (K - 1) \log \frac{2\epsilon}{\epsilon}$$

$$= K - 1.$$

Substituting this into Equation 7 we obtain that

$$\frac{\mathcal{BR}^*(T)}{\epsilon T} \geq \frac{1}{2} \left[ 1 - \frac{1 + 2\bar{A}\epsilon^2 T}{K - 1} \right].$$

Now, we select $\epsilon$ to maximize such that our lower bound will be the tightest. In particular, we choose $\epsilon = \sqrt{\frac{K-1}{6T\bar{A}}}$ which yields the following lower bound,

$$\mathcal{BR}^*(T) \geq \frac{1}{2} \sqrt{\frac{T(K-1)}{6\bar{A}}} \left[ \frac{2}{3} - \frac{1}{K-1} \right].$$

This concludes our proof. $\square$

The results for multi-armed bandits and tabular MDPs are shown by substituting $|\Pi| = K$ and $HSK$ respectively.

**Proposition C.2.** *For any algorithm, there exists a Bayesian decision-making task with $\Pi \subseteq \mathbb{B}_2^d$, where the reward means are a linear function of the decisions, such that the regret can be lower bounded by $\Omega(\sqrt{dT})$.*

*Proof.* Similarly to Proposition C.1, we have $\rho(x, y) = c \|x - y\|_1$ for some positive constant $c > 0$. We focus on the set $\mathbb{B}_2^d$ and note that it is a parametric set. Furthermore, using Lemma B.1 we have that

$$\log \mathcal{M}^{\text{loc}}(\rho, \mathbb{B}_2^d, \epsilon) \geq d.$$

Substituting this into Equation 8 we obtain that

$$\frac{\mathcal{BR}^*(T)}{\epsilon T} \geq \frac{1}{2} \left[ 1 - \frac{1 + 2\bar{A}\epsilon^2 T}{d} \right].$$

Now, we select $\epsilon$ to maximize such that our lower bound will be the tightest. In particular, we choose $\epsilon = \sqrt{\frac{d}{6T\bar{A}}}$ which yields the following lower bound,

$$\mathcal{BR}^*(T) \geq \frac{1}{2} \sqrt{\frac{Td}{6\bar{A}}} \left[ \frac{2}{3} - \frac{1}{d} \right].$$

This concludes our proof. $\qquad\square$

In the Lipshitz bandit setting, $\Pi$ is some metric space with metric $\rho$ and $\mathcal{M} = \mathcal{M}_{\mathcal{F}}$, where

$$\mathcal{F} = \{ f : \Pi \to [0, 1] | f \text{ is 1-Lipschitz w.r.t } \rho \}.$$

Unlike the previous settings, we note that the decision set is not parametric this time. We now prove a lower bound for this setting.

**Proposition C.3.** *The regret lower bound for the Lipschitz bandit is $\Omega\left( T^{\frac{d+1}{d+2}} \right)$, where $d$ is chosen such that*

$$c_1 \leq \log \mathcal{N}(\rho, \epsilon, \Pi) \cdot \epsilon^d \leq c_2$$

*Proof.* From the assumption, the covering and packing number can now be bounded by

$$\log \mathcal{N}\left( \rho, \sqrt{\frac{2}{\delta}}, \Pi \right) \leq c_2 \left( \frac{2}{\delta} \right)^{d/2}$$

$$\log \mathcal{M}(\rho, \epsilon, \Pi) \geq c_1 \epsilon^{-d}.$$

Substituting this into Equation 9,

$$\frac{\mathcal{BR}^*(T)}{\epsilon T} \geq \frac{1}{2} \left[ 1 - \left( 1 + c_2 \left( \frac{2}{\delta} \right)^{d/2} + T\delta \right) \frac{\epsilon^d}{c_1} \right].$$

We choose $\delta = T^{\frac{2}{d+2}} c_2^{-\frac{2}{d+2}} 2^{-\frac{d}{d+2}}$. This yields that

$$\frac{\mathcal{BR}^*(T)}{\epsilon T} \geq \frac{1}{2} \left[ 1 - \left( 1 + 2(2c_2)^{-\frac{d}{d+2}} T^{\frac{d}{d+2}} \right) \frac{\epsilon^d}{c_1} \right].$$

Selecting $\epsilon^d = \frac{1}{2} \left[ \frac{c_1}{\left( 1 + (2c_2)^{-\frac{d}{d+2}} T^{\frac{d}{d+2}} \right)} \right]$ yields

$$\frac{\mathcal{BR}^*(T)}{\epsilon T} \geq \frac{1}{4}$$

Now, $\epsilon T \gtrsim T^{\frac{d+1}{d+2}}$ so we have that

$$\mathcal{BR}^*(T) \gtrsim T^{\frac{d+1}{d+2}}.$$

$\qquad\square$

We now present the following theorem, which shows that the worst-case Bayesian regret is equal to the mini-max frequentist regret.

**Theorem C.4** (Theorem 1 of Lattimore & Szepesvári (2019)).

$$\inf_{\{\pi_i\}_{i=1}^T} \sup_{r_1,\ldots,r_t} \max_{\pi^* \in \Pi} \mathbb{E}\left[\sum_{t=1}^T (r_t(\pi^*) - r_t(\pi_t))\right] = \sup_{\nu} \inf_{\{\pi_i\}_{i=1}^T} \max_{\pi^* \in \Pi} \mathbb{E}\left[\sum_{t=1}^T (r_t(\pi^*) - r_t(\pi_t))\right]$$

This theorem then states that, without any constraints on the prior, the worst-case Bayesian regret is equal to the mini-max regret. This means that any lower bound derived for the Bayesian regret can also be applied to the frequentist one.

### C.4 ADDITIONAL EXPLANATION

We now provide an additional explanation regarding Proposition 3.4. In particular, we explain why we had to select a local packing set for parametric decision spaces $\Pi$ and why we focused on the simplex of the decision set for finite decision spaces.

**Selecting $\Delta(\Pi)$ over $\Pi$.** This explanation is rather straightforward. We see that by selecting $\Pi$ we have that $\mathcal{M}^{\mathrm{loc}}(\rho, \Pi, \epsilon) = 1$ for any value of $\epsilon$ which makes the lower bound null.

**Selecting a local-packing set for parametric $\Pi$.** This decision stems from the improved upper bound of $I(V; \mathcal{H}_T)$ for parametric decision space, which is found in Theorem 3.3. Using the other upper bound for $I(V; \mathcal{H}_T)$ simply results in a sub-optimal lower bound. Additional details can be found in Yang & Barron (1999).

## D PROOFS FOR SECTION 4

We begin by stating and proving the following Lemma.

**Lemma D.1.** *Under the assumptions and notations of Proposition 3.2, the constraint of $I(\pi^*; \mathcal{H}_T) \leq R \leq \log K$, and the regularity Assumption 3.1, the following holds.*

$$I(V; \mathcal{H}_T) \leq 2\bar{A}T \frac{R}{\log K} \epsilon^2.$$

*Proof.* The proof follows a similar analysis to the proof of Theorem 3.3, which is provided in Yang & Barron (1999). Since we focus on a parametric set, we consider a local packing set with a cardinality of $\mathcal{M}^{\mathrm{loc}}(\rho, \epsilon, \Delta(\Pi))$, which we denote by $E$. $\tilde{E}$ is the set contained by $E$ under the mutual information constraint,

$$\tilde{E} = \{p \in E : I(\pi^*; \mathcal{H}_T) \leq R\}.$$

We now have

$$\begin{aligned}
I(V; \mathcal{H}_T) &= \sum_{t=1}^T I(V; C_{t+1}, O_{t+1}, \ldots, C_T, O_T \mid \mathcal{H}_t) \\
&= \sum_{t=1}^T D_{\mathrm{KL}}\left(P^{V|\mathcal{H}_T} \parallel P^{V|\mathcal{H}_t}\right) \\
&\leq T \max_{P_1, P_2 \in \tilde{E}} D_{\mathrm{KL}}\left(P_1 \parallel P_2\right) \\
&\leq T \frac{R}{\log K} \max_{P_1, P_2 \in E} D_{\mathrm{KL}}\left(P_1 \parallel P_2\right) \\
&\leq T \frac{R}{\log K} \bar{A} \max_{P_1, P_2 \in E} \rho(P_1, P_2)^2 \\
&\leq T \frac{R}{\log K} \bar{A} \epsilon^2.
\end{aligned}$$

where the second inequality follows from a scaling argument, and the last follows from the fact that $E$ is a local packing set. $\qquad \square$

As a corollary from Lemma D.1, we obtain the following.

**Corollary D.2** (Proposition 4.2). *Let there be a Bayesian interactive decision-making problem as defined in Section 2, with a finite decision space $\Pi$, where $|\Pi| = K$. Then,*

$$\frac{\mathcal{BR}^*(T)}{T\epsilon} \geq \frac{1}{2}\left[1 - \frac{2\bar{A}\epsilon^2 T \frac{R}{\log K} + 1}{\log \mathcal{M}^{\mathrm{loc}}(\rho, \Delta(\Pi), \epsilon)}\right]$$

*Furthermore, if the decision-making problem is a part of a parametric set, then*

$$\frac{\mathcal{BR}^*(T)}{T\epsilon} \geq \frac{1}{2}\left[1 - \frac{2\bar{A}\epsilon^2 T \frac{R}{\log K} + 1}{\log \mathcal{M}^{\mathrm{loc}}(\rho, \Pi, \epsilon)}\right]$$

*Proof.* The proof follows immediately by utilizing Lemma D.1 and using the fact that in this scenario, $\rho(x, y) = \|x - y\|_1$, similarly to the proof of Proposition C.1. $\qquad\square$

### D.1 PROOF OF PROPOSITION 4.1

*Proof.* We start by considering $R = 0$. From Lemma 4.2,

$$\mathcal{BR}^*(T) \geq \frac{\epsilon T}{2}\left(1 - \frac{1}{K}\right)$$

for any $\epsilon > 0$ such that lower bounds the difference between two different decisions. Since the maximal difference between two decisions is 1, we set $\epsilon = 1$ and obtain the linear lower bound,

$$\mathcal{BR}^*(T) \geq \frac{T}{2}\left(1 - \frac{1}{K}\right).$$

We now consider $R > 0$. We denote the polytope space with the mutual information constraint by $\Delta_R$. Similarly to the proof of Proposition C.1,

$$\log \mathcal{M}^{\mathrm{loc}}(\rho, \Delta(\Pi), \epsilon) \geq (K - 1).$$

Substituting this into Equation 10,

$$\frac{\mathcal{BR}^*(T)}{T\epsilon} \geq \frac{1}{2}\left[1 - \frac{2T\bar{A}\frac{R}{\log K}\epsilon^2 + 1}{(K-1)}\right].$$

Selecting $\epsilon = \sqrt{\frac{(K-1)\log K}{6RT\bar{A}}}$ we now have that

$$\mathcal{BR}^*(T) \geq \frac{1}{2}\sqrt{\frac{TK\log K}{6R\bar{A}}}\left(\frac{2}{3} - \frac{1}{K-1}\right)$$

We present the proof for the linear case. Now,

$$\log \mathcal{M}^{\mathrm{loc}}(\rho, \Pi, \epsilon) \geq d.$$

So now we have

$$\frac{\mathcal{BR}^*(T)}{T\epsilon} \geq \frac{1}{2}\left[1 - \frac{2T\frac{R}{\log K}\bar{A}\epsilon^2 + 1}{d}\right].$$

Selecting $\epsilon = \sqrt{\frac{d\log K}{6R\bar{A}T}}$ we now have that

$$\mathcal{BR}^*(T) \geq \frac{1}{2}\sqrt{\frac{Td\log K}{6R\bar{A}}}\left(\frac{2}{3} - \frac{1}{d}\right)$$

$\qquad\square$

Proposition 4.1 can then be applied directly on the MAB setting.

**Corollary D.3.** *Consider a Bayesian MAB with $K$ arms. If $I(\pi^*; \mathcal{H}_T) = 0$ then the worst-case regret is linear. Otherwise, for $I(\pi^*; \mathcal{H}_T) \geq 0$, the worst-case regret can be lower-bounded by $\Omega\left(\sqrt{\frac{TK\log K}{I(\pi^*; \mathcal{H}_T)}}\right)$. Additionally, if $\Pi = \mathbb{B}_2^d$ and the reward means are linear functions of the decisions, we have a lower bound of $\Omega\left(\sqrt{\frac{Td\log K}{I(\pi^*; \mathcal{H}_T)}}\right)$.*

## D.2 PROOF OF PROPOSITION 4.3

*Proof.* We consider the following two algorithms. The first is Thompson sampling, which means that, $p_t = P^{\pi^* | \mathcal{H}_\sqcup}$. The second algorithm is Thompson sampling with the constrained observations $\tilde{O}_t$ such that $I(\pi^*; \tilde{\mathcal{H}}_T) = D_{\mathrm{KL}} \left( P^{\pi^* | \tilde{\mathcal{H}}_T} \parallel P^{\pi^*} \right) = R$, where $\tilde{\mathcal{H}}_t$ is the history with constrained observations. Now, we define the information ratio for both the normal and constrained variants of Thompson sampling:

$$\rho_t = \frac{\mathbb{E}[r_M(\pi^*(C_t), C_t) - r_M(p_t, C_t)]^2}{I(\pi^*; O_{t+1} \mid \mathcal{H}_t)} \quad ; \quad \tilde{\rho}_t = \frac{\mathbb{E}[r_M(\pi^*(C_t), C_t) - r_M(p_t, C_t)]^2}{I(\pi^*; \tilde{O}_{t+1} \mid \tilde{\mathcal{H}}_t)}.$$

We see that

$$\begin{aligned} \frac{\rho_t}{\tilde{\rho}_t} &= \frac{I(\pi^*; \tilde{O}_{t+1} \mid \tilde{\mathcal{H}}_t)}{I(\pi^*; O_{t+1} \mid \mathcal{H}_t)} \\ &\geq \frac{I(\pi^*; \tilde{O}_{t+1} \mid \tilde{\mathcal{H}}_t)}{\log K} \end{aligned}$$

where the inequality follows from $I(\pi^*; O_{t+1} \mid \mathcal{H}_t) \leq \log K$. Thus,

$$\begin{aligned} \max_P \max_t \tilde{\rho}_t &\leq \max_P \max_t \frac{\log K}{D_{\mathrm{KL}} \left( P^{\pi^* | (\tilde{O}_t, \tilde{\mathcal{H}}_t)} \parallel P^{\pi^* | \tilde{\mathcal{H}}_t} \right)} \rho_t \\ &\leq \frac{\log K}{R} \max_t \rho_t \\ &\leq \frac{K \log K}{2R} \end{aligned}$$

Where the third inequality follows by the information ratio upper bound presented by Russo & Van Roy (2016). If the MAB is linear then the information ratio $\rho_t$ can be upper-bounded by $\frac{d}{2}$. Then, we have the following upper bound,

$$\max_P \max_t \tilde{\rho}_t \leq \frac{d \log K}{2R}$$

Now, by the analysis done by Russo & Van Roy (2016), we know that if $\max_t \tilde{\rho}_t \leq \tilde{\rho}$ for all $t$, then the Bayesian regret can be upper bounded by

$$\mathcal{BR}^*(T) \leq \sqrt{\tilde{\rho} T \log K}$$

Thus, we have for $K$-MAB:

$$\mathcal{BR}^*(T) \leq \log K \sqrt{\frac{KT}{2R}}.$$

And for linear bandits:

$$\mathcal{BR}^*(T) \leq \log K \sqrt{\frac{dT}{2R}}.$$

$\square$

## D.3 PROOF OF PROPOSITION 4.5

We begin by stating and proving the following Lemma.

**Lemma D.4.** *Let $P_1, P_2$ be two priors such that $H(P_1^{\pi^*}) = R \leq \log K$. Also let $\psi, \phi$ be two decisions such that $D_{\mathrm{KL}} \left( \psi \parallel P_1^{\pi^*} \right) \geq c \bar{A} R$ and $D_{\mathrm{KL}} \left( \phi \parallel P_2^{\pi^*} \right) \leq \log K$. Then,*

$$\frac{\mathbb{E}_{P_1}[\mu_M(\pi^*, C) - \mu_M(\psi, C)]}{\mathbb{E}_{P_2}[\mu_M(\pi^*, C) - \mu_M(\phi, C)]} \geq 2 \sqrt{c \frac{R}{\log K}}$$

*Proof.* By the regularity assumption, we know that

$$(\mathbb{E}_{P_1}[\mu_M(\pi^*, C) - \mu_M(\psi, C)])^2 \geq \frac{cR}{2}.$$

From Pinsker's inequality,

$$(\mathbb{E}_{P_2}[\mu_M(\pi^*, C) - \mu_M(\phi, C)])^2 \leq 2D_{\mathrm{KL}}\left(\phi \parallel P_2^{\pi^*}\right) \leq \log K.$$

Dividing these two inequalities yields

$$\frac{(\mathbb{E}_{P_1}[\mu_M(\pi^*, C) - \mu_M(\psi, C)])^2}{(\mathbb{E}_{P_2}[\mu_M(\pi^*, C) - \mu_M(\phi, C)])^2} \geq 4c\frac{R}{\log K}$$

which concludes our proof. $\qquad\square$

Now, we prove the following Proposition.

**Proposition D.5.** *Let $P$ be a prior $P$ with $H(\pi^*) = R \leq \log K$. If Assumption 4.1 holds, then*

$$\mathcal{BR}_P(T; \{p_t\}) \geq 2\sqrt{\frac{cR}{\log K}}\mathcal{BR}^*(T). \tag{15}$$

*Proof.* We see that

$$\mathcal{BR}_P(T; p_t) = \sum_{t=1}^{T} \mathbb{E}_P[\mu_M(\pi^*, C_t) - \mu_M(p_t, C_t)]$$

$$\geq 2\sqrt{c\frac{R}{\log K}} \sum_{t=1}^{T} \mathbb{E}_\nu[\mu_M(\pi^*, C_t) - \mu_M(p_t', C_t)]$$

for any prior $\nu$ and stochastic decisions $p_t'$ such that $D_{\mathrm{KL}}\left(p_t' \parallel \nu^{\pi^*}\right) \leq \log K$. So we can take the worst-case regret,

$$\mathcal{BR}_P(T; p_t) \geq 2\sqrt{c\frac{R}{\log K}} \max_\nu \inf_{\{p_t'\}: D_{\mathrm{KL}}\left(p_t' \parallel \nu^{\pi^*}\right) \leq \log K} \sum_{t=1}^{T} \mathbb{E}_\nu[\mu_M(\pi^*, C_t) - \mu_M(p_t', C_t)]$$

$$\geq 2\sqrt{c\frac{R}{\log K}} \max_\nu \inf_{\{p_t'\}} \sum_{t=1}^{T} \mathbb{E}_\nu[\mu_M(\pi^*, C_t) - \mu_M(p_t', C_t)]$$

$$= 2\sqrt{c\frac{R}{\log K}}\mathcal{BR}^*(T)$$

where the second inequality follows by removing a constraint on the available decisions. $\qquad\square$

This results in the following Corollary.

**Corollary D.6** (Proposition 4.5). *Let there be a $K$-MAB. Then for any algorithm such that $D_{\mathrm{KL}}\left(p_t \parallel P^{\pi^*}\right) \geq c\bar{A}H(\pi^*)$, the regret can be lower-bounded by $\Omega\left(\sqrt{\frac{KH(\pi^*)T}{\log K}}\right)$. Additionally, if the MAB is linear, then the regret can be lower bounded by $\Omega\left(\sqrt{\frac{dH(\pi^*)T}{\log K}}\right)$.*

We now explain why in scenarios where the optimal decision is unique, an algorithm that selects the optimal arm with high probability obeys Assumption 4.1.

**Proposition D.7.** *Let there be an algorithm such that $p_t$ selects the optimal decision $\pi^*$ with probability $1 - \delta$. Then,*

$$D_{\mathrm{KL}}\left(p_t \parallel P^{\pi^*}\right) \geq (1 - \delta)H(\pi^*) - c_\delta$$

*where $c_\delta \to 0$ as $\delta \to 0$.*

*Proof.* We can write that

$$D_{\mathrm{KL}}\left(p_t \parallel P^{\pi^*}\right) = \mathbb{E}\left[\sum_\pi p_t(\pi) \log\left(\frac{p_t(\pi)}{P^{\pi^*}(\pi)}\right)\right]$$

$$\geq \mathbb{E}_P\left[(1-\delta)\log\left(\frac{1-\delta}{P^{\pi^*}(\pi^*)}\right)\right]$$

$$= (1-\delta)H(\pi^*) - (1-\delta)\log\left(\frac{1}{1-\delta}\right)$$

where the second inequality follows from the fact that $p_t(\pi^*) \geq 1 - \delta$. $\qquad\square$

From this, we can conclude that any algorithm whose stochastic decision at round $t$ converges to the optimal decision obeys the assumption.

# E  BANDIT EXPERIMENTS

## E.1  ADDITIONAL DETAILS

The mutual information for the experiments in this setting is estimated using the KL divergence between the output probability distribution of every algorithm with the prior probability, $D_{\mathrm{KL}}\left(p_t \parallel P^{\pi^*}\right)$. This is an estimator of the mutual information, similar to the one used by Seldin et al. (2011). The mutual information was then calculated according to $D_{\mathrm{KL}}\left(p_t \parallel P^{\pi^*}\right)$, where $p_t$ is the real posterior. In our problem, the posterior update is done in the following manner.

$$P(\pi^* = i \mid \mathcal{H}_t) = \frac{P(\mathcal{H}_t \mid i)P(i)}{P(\mathcal{H}_t)}$$

$$= \frac{P(\mathcal{H}_t \mid i)P(i)}{\sum_j P(\mathcal{H}_t \mid i)P(i)}$$

where

$$P(\mathcal{H}_t \mid i) = \left(\frac{1+\epsilon}{2}\right)^{N_{s,i}}\left(\frac{1-\epsilon}{2}\right)^{N_{f,i}}\prod_{j\neq i}\left[\left(\frac{1-\epsilon}{2}\right)^{N_{s,j}}\left(\frac{1+\epsilon}{2}\right)^{N_{f,j}}\right],$$

and $N_{s_i}$ is the number of successful pulls (pulls that received reward 1) of arm $i$. $N_{f,i}$ is the number of failed pulls of arm $i$.

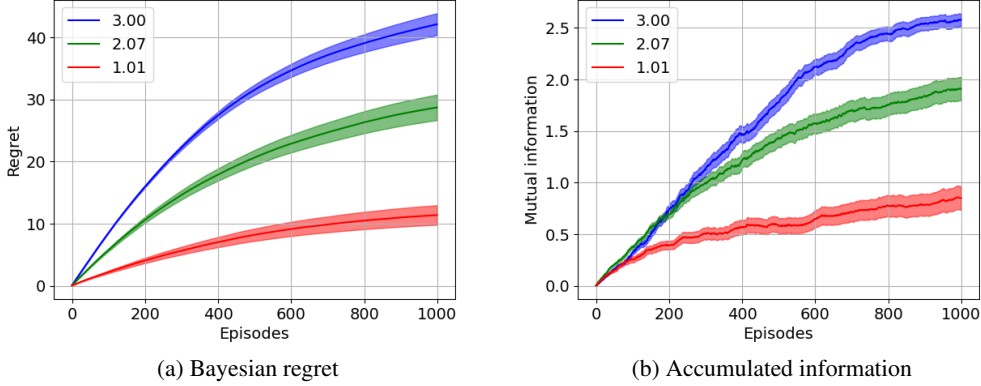

(a) Bayesian regret  (b) Accumulated information

Figure 3: (a) The Bayesian regret and (b) the accumulated information in bits for Thompson sampling under three different priors over the same bandit structure. The entropy of each prior is described in the legend. The shadowed areas correspond to 2-sigma error bars.

Figure 4: The prompting done for the LLMs in our experiments. The question is preceded by a question prompt followed by "Answer the following question:". Following this, the options are presented. <Question prompt> and <End prompt> are both replaced by a different prompt for every model.

In both bandit experiments, we used 200 random seeds. Each random seed considers both the randomness in the problem generated (since the bandit problem is generated via a prior) and the randomness during the algorithm's execution. We then reported in each graph the standard error for both the regret and mutual information, which is given by $\text{err} = \frac{2\sigma}{\sqrt{n}}$ where $\sigma$ is the standard deviation and $n$ is the number of seeds.

## E.2 ADDITIONAL EXPERIMENTS

We use the same bandit setting as in Section 5. In this experiment, we compare the regret of Thompson sampling under three $H(\pi^*)$ values. We use the same model space and number of arms. Results are presented in Figure 3. As we can see, when less information can be accumulated, less regret is suffered.

## F  MCQA WITH LLM EXPERIMENT

### F.1  ADDITIONAL DETAILS

In this experiment, we used a small LLM and a large LLM in every experiment. The small LLM was one of the following: Mistral 7B (Jiang et al., 2023), Gemma 2B (Team et al., 2024) and Falcon 7B (Almazrouei et al., 2023). For the large LLM we used Mixtral 8x7B (Jiang et al., 2024), Llama3-70B (AI@Meta, 2024) and Gemma 7B (Team et al., 2024). 4-bit quantization (Dettmers & Zettlemoyer, 2023) and flash attention (Dao et al., 2022) were applied for all models (excluding Falcon 7B, for which flash attention was not applied). In every round, we queried the LLM with a prompt, which is described later on. An example of the prompt used is provided in Figure 4. Using the LLM's output we obtain a probability distribution over the tokens of the possible answers [A,B,C,D]. We then selected the answer according to this distribution and received feedback on whether the answer was correct, which was our reward. We also see that performance improvement is not clear when we use Mistral for the small model and Gemma 7B for the large one. This is because they are similar in size, while for the rest of the experiments, the larger LLM has significantly more parameters.

Table 4: Regret for 10 different seeds for random and bits-based policies in LLM selection for MCQA tasks. Both policies were allowed to query the large LLM for only 50% of the episodes. The top row corresponds with the large LLM and the left column with the small one.

|  | **Random** | | | **Bits-Based** | | |
|---|---|---|---|---|---|---|
|  | Mixtral | Llama3 | Gemma 7B | Mixtral | Llama3 | Gemma 7B |
| Mistral | $38.7 \pm 0.3$ | $47.0 \pm 0.5$ | $40.4 \pm 0.3$ | $\mathbf{36 \pm 1}$ | $46 \pm 1$ | $40 \pm 1$ |
| Gemma 2B | $44.5 \pm 0.7$ | $54.1 \pm 0.6$ | $39.3 \pm 0.3$ | $\mathbf{38 \pm 2}$ | $46 \pm 2$ | $\mathbf{35.3 \pm 0.9}$ |
| Falcon 7B | $44.0 \pm 0.3$ | $52.2 \pm 0.4$ | $45.5 \pm 0.3$ | $\mathbf{35.0 \pm 0.8}$ | $44 \pm 1$ | $39 \pm 1$ |

Table 5: Regret for 10 different seeds for random and bits-based policies in LLM selection for MCQA tasks. Both policies were allowed to query the large LLM for only 70% of the episodes. The top row corresponds with the large LLM and the left column with the small one.

| | Random | | | Bits-Based | | |
|---|---|---|---|---|---|---|
| | Mixtral | Llama3 | Gemma 7B | Mixtral | Llama3 | Gemma 7B |
| Mistral | $30.5 \pm 0.4$ | $33.8 \pm 0.4$ | $34.8 \pm 0.3$ | $\mathbf{26.6 \pm 0.8}$ | $\mathbf{31 \pm 1}$ | $\mathbf{33.0 \pm 0.8}$ |
| Gemma 2B | $35.6 \pm 0.8$ | $40.1 \pm 0.6$ | $32.9 \pm 0.3$ | $\mathbf{32 \pm 1}$ | $\mathbf{36 \pm 2}$ | $\mathbf{30 \pm 1}$ |
| Falcon 7B | $34.0 \pm 0.3$ | $37.0 \pm 0.3$ | $37.9 \pm 0.2$ | $\mathbf{27.0 \pm 0.7}$ | $\mathbf{31 \pm 1}$ | $\mathbf{33 \pm 1}$ |

We repeated this process for 10 different question seeds. Table 2 reports the regret after 200 steps for all combinations of LLMs across all 10 seeds. Additional results are presented in Tables 4 and 5. Both tables compare the random selection method and the bits-based one for different LLMs. Table 4 compares the regret for 50% deployment rate of the large LLM and Table 5 compares the regret for 70% deployment rate of the large LLM. We conclude from both tables that the effect of the bits-based policy increases with the deployment rate of the large LLM. Furthermore, we also see that bits-based selection becomes more significant, as the performance gap between the small and large LLM increases as well.

## F.2 COMPUTE SPECIFICATIONS

The bandit experiments are performed on a CPU and do not require special compute workers. The experiments using Mistral 7B, Gemma 2B, Gemma 7B and Falcon 7B were evaluated using a machine with RTX4090. The experiments using Mixtral 8x7B and Llama3-70B were performed using a machine with 3xA40.

## F.3 DATASET AND MODELS LICENSE

The MMLU dataset (Hendrycks et al., 2021b;a) is available under the *MIT license*. Mistral 7B (Jiang et al., 2023), Mixtral 8x7B (Jiang et al., 2024) and Falcon 7B (Almazrouei et al., 2023) are available under the *Apache license 2.0*. Gemma 2B and 7B (Team et al., 2024) are available under the *Gemma license*. Llama3-70B (AI@Meta, 2024) is available under the *Llama license*.

## F.4 LLM PROMPTS

We now describe the different prompts for every LLM.

### F.4.1 MISTRAL AND MIXTRAL

> **Prompt for generating answers for Mistral 7B and Mixtral 8x7B**
>
> You will answer the following question using one of the following letters, A, B, C, or D. Do not explain or describe the answer. You are given the following question:
> <Question>
> The possible answers are:
> A. <Option A>
> B. <Option B>
> C. <Option C>
> D. <Option D>
> Please output only the letter corresponding with the correct answer - A, B, C or D. Don't explain or describe the answer.
> Your answer:

### F.4.2 LLAMA3

---

**Prompt for generating answers for Llama3 8B / 70B**

You are a bot that only outputs one of the following letters - A, B, C or D. You are designed to answer multiple choice questions. Do not explain or describe the answer.
Question: You are given the following question:
<Question>
The possible answers are:
A. <Option A>
B. <Option B>
C. <Option C>
D. <Option D>
You must output only the letter corresponding with the correct answer - A, B, C or D. Don't explain or describe the answer.
Output:

---

### F.4.3 FALCON

---

**Prompt for generating answers for Falcon 7B**

You are an AI assistant designed to answer multiple choice questions. You will answer the following question using one of the following letters, A, B, C, or D. Do not explain or describe the answer. You are given the following question:
<Question>
The possible answers are:
A. <Option A>
B. <Option B>
C. <Option C>
D. <Option D>
Please output only the letter corresponding with the correct answer - A, B, C or D. Don't explain or describe the answer.
Your answer:

---

### F.4.4 GEMMA 2B / 7B

---

**Prompt for generating answers for Gemma 2B / 7B**

<start_of_turn>user
You will answer the following question using one of the following letters, A, B, C, or D. Do not explain or describe the answer. You are given the following question:
<Question>
The possible answers are:
A. <Option A>
B. <Option B>
C. <Option C>
D. <Option D>
Please output only the letter corresponding with the correct answer - A, B, C or D. Don't explain or describe the answer.<end_of_turn>
<start_of_turn>model

---

