# OpenReview forum: "On Bits and Bandits: Quantifying the Regret-Information Trade-off"
_ICLR.cc/2025/Conference — ICLR 2025 Poster_

### Official Review · Reviewer_z6VF · 2024-11-03

**Soundness:** 3
**Presentation:** 3
**Contribution:** 2
**Rating:** 6
**Confidence:** 3

**Summary:**

The submission studies the information lower bounds of bandits. By relating the mutual information to KL divergence and entropy, the submission rephrases regret bounds in terms of information bits. The new form of the bounds allows a learning agent to acquire and accumulate additional knowledge through active queries (as opposed to passive observations in the previous setting). The main results are summarized in Table 1, consisting of Theorem 3.4, Proposition 4.1 and Proposition 4.5. In the experiments, the advantage of information accumulation is verified in a simulation (Figure 1). Then, a query strategy is proposed and tested in MCQA.

**Strengths:**

- The submission points out that using Fano, one can introduce the mutual information into the regret bound. The connection between the mutual information and the accumulated knowledge bits (R) provides a means to analyze the effect of the knowledge bits (R) on the regret bound.
- Moreover, Prop. 4.5 provides an entropy-dependent Bayesian regret lower bound, which is listed in the last entry of Table 1.
- The advantage of accumulating information in bits is experimentally justified (Figure 2).
- A bits-based query policy illustrates the advantage of quantifying knowledge in bits and searching for a query that will bring an abundant increase in the knowledge accumulation.

**Weaknesses:**

- (1) Combining Prop. 4.1 and 4.3, the submission provides a range of bits required to achieve a given level of regret. Note that the range has a $\sqrt{logK}$ gap.
- (2) The proof sketches of the main results (e.g., Proposition 3.2, Theorem 3.4, Proposition 4.1, and the others) are plain. It seems that packing and covering are standard techniques in analysis. Could you please elaborate on the technical challenges and the corresponding contributions in the sketches?

**Questions:**

- (3) The first column in Table 2 resembles a budgeted setting. Can there be an autonomous scenario? For instance, let the learning agent decide the proportion of queries.

---

> ### Author Response · Authors · 2024-11-17
>
> We want to thank the reviewer for the detailed analysis and feedback. Below is a response addressing your concerns:
>
> (W1) We acknowledge this gap and agree it would be interesting to close it. Still, in settings where $K$ is small, as in many MAB problems, this factor is not too significant. In particular, many works that study problem-independent regret ignore log-factors. Moreover, the bound still gives non-trivial insights, especially concerning the relationship between information and regret.
>
> (W2) The techniques in this paper include the adaptation of the method presented by [Yang and Barron, 1999] to an online setting. This includes utilizing covering and packing numbers to introduce a regret lower bound. One contribution we introduce is the selection of $\Phi$ for different policy classes, allowing us to obtain tighter lower bounds for a wide variety of decision spaces. Another technical contribution is modifying this method to handle problems with information constraints.
>
> (Q3) Allowing the agent to adaptively determine when to query external sources for information is a very interesting setup. While it is outside the scope of this work, we will hopefully examine it in future work.

---

> > ### Comment · Reviewer_z6VF · 2024-11-26
> >
> > Thank you for the feedback. I will keep my score.

---

### Official Review · Reviewer_smVV · 2024-11-04

**Soundness:** 3
**Presentation:** 1
**Contribution:** 3
**Rating:** 6
**Confidence:** 3

**Summary:**

The paper studies a general sequential decision-making framework from a Bayesian perspective. Within this framework, it is intuitive that the more information the agent accumulates, the lower the resulting regret. The goal of the paper is to formalize that intuition. The paper does the following:
1. Develops new information-theoretic methods for analyzing sequential decision-making algorithms
2. Uses those methods to recover existing lower bounds for a range of sequential decision-making problems, such as standard multi-armed bandits, tabular RL, Lipschitz bandits, etc (Table 1).
3. Obtains lower and upper regret bounds which depend explicitly on the number of bits of information accumulated.
4. Runs a question-answering LLM experiment inspired by the above results.

**Strengths:**

I think this work has many of the ingredients of a strong conceptual paper. The authors identify a conceptual phenomenon which spans many mathematical models, formalize that phenomenon, and develop a method which can analyze this phenomenon simultaneously in all of those models.

Although the LLM experiment initially felt out of place to me, I actually think it provides a nice complement to the theoretical results (although the theoretical results certainly remain the primary contribution).

Overall, I think the ceiling for this paper is quite high.

**Weaknesses:**

I have serious concerns about the presentation. Although the conceptual idea behind the paper is intuitive, it took me a while to make sense of the technical content of the paper. I think there are two issues:
1. Confusing writing and non-standard terminology.
2. Lack of explanation of the technical statements.
I have provided a non-exhaustive list of examples below.

Although I am not an expert in information-theoretic methods, I am quite familiar with bandits, RL, and Bayesian regret, so more casual readers may struggle even more than me. If the paper purports to elucidate the intuitive tradeoff between information and regret, but the technical results are not accessible to readers, then I believe the impact of the paper will be limited.

I also think the LLM experiments could be improved by including baselines of always querying/never querying the large LLM. Table 2 suggests that with the query cost of 0.1, always querying the large LLM might actually be the optimal policy. To me, this suggests that a larger query cost is needed and calls into question the significance of the evaluation.

Overall, although I think the paper has many merits, I lean towards rejection so that these issues can be addressed, hopefully resulting in a strong final paper.

_Writing issues_
1. I found it a bit hard to make sense of Section 1.1 (“Contributions”) without at least informally defining the model. It would also be useful to link to the theorems/sections corresponding to each of the results.
2. Some of the terminology and notation is a bit confusing. Normally $\pi \in \Pi$ denotes a policy, but here it denotes a “decision” (basically an action). Instead, $\phi \in \Delta(\Pi)$ is called a policy, which seems like it should just be called a randomized decision/action. Furthermore, $p_t: \mathcal{C} \to \Delta(\Pi)$ is _also_ called a policy, which is more in line with the normal usage of “policy”. And $\pi^*$ is also a function from $\mathcal{C} \to \Delta(\Pi)$, which gives it a different type than $\pi$. I have also never before seen the term “epsilon-local set” used to describe epsilon-balls. I would suggest better aligning terminology and notation with the literature.
3. In Example 2.1, is there a reason that you use Bernoulli rewards instead of general rewards? Does your model not cover contextual MAB with general rewards.
4. Lines 197 - 215: I assume the rho separation assumption is for policies in $\Phi$, not for policies in $\Delta(\Pi)$? If it is supposed to be $\Delta(\Pi)$, that seems like a very strong assumption about the structure of the decision space.

_Lack of interpretation/explanation_
1. I understand that Yang and Barron also make Assumption 3.1, but it seems pretty unintuitive to me, and I would have appreciated some explanation.
2. Theorem 3.4, especially (9), is a bit hard to make sense of. Could you provide an interpretation for this expression?

_Minor issues_
1. Line 41: resource allocation is much broader than the specific routing problem you describe. Consider something like “One example of a resource allocation problem is route a process…” The flow in this section also feels a bit weird since you never bring up resource allocation again in the paper. Consider omitting either the resource allocation or online game example and using a single running example?
2. Since Section 4 also includes upper bounds, should the title be “Information-theoretical regret upper and lower bounds?”
3. Table 2 caption: the “Appendix ??” reference is broken

**Questions:**

I don’t have further questions beyond what I’ve written above in “Weaknesses”.

---

> ### Author Response · Authors · 2024-11-17
>
> We appreciate the reviewer’s feedback and have addressed the issues raised. We uploaded a revised version of the paper, where the modifications made are highlighted. Below is a response addressing your concerns:
>
> Regarding your concerns over the LLM experiments, querying the large LLM only is not the optimal policy. This is due to the 0.1 penalty, making the small LLM better than the large one for some questions. However, choosing the large LLM only will be better than both policies – since our focus is when queries are constrained. We have also added the baselines of using only one model to Table 2.
>
> _Writing issues_
>
> 1. We have modified the introduction to briefly introduce the setting before the contributions. We also added references to the appropriate Theorems / Propositions in this section.
> 2. In our denoting of $\pi$ as a decision and $\Pi$ as a decision set, we will further emphasize that we follow the notations of papers on decision-making settings, e.g. that of (Xu and Zeevi 2023, Foster et. al. 2023). $\pi^*$ is a different type than $\pi$ as it is the optimal decision for every context. However, we changed $\pi^* : \mathcal{C} \to \Pi$ to $\pi^*_{M,C} \in \Pi$ to be less confusing. We also changed our notion of $p_t \in \Delta(\Pi)$ from policy to stochastic decision. We also changed the term $\epsilon$-local set to $\epsilon$-ball to better align with existing terminology.
> 3. Bernoulli rewards were chosen mainly for simplicity of exposition. Our setting also covers general rewards as well. We added this comment to the paper as well.
> 4. While Proposition 3.2 requires this assumption to hold only for the decisions in $\Phi$, for Theorem 3.4 to hold $\Phi$ needs to be a packing set. Hence we assume separation for all stochastic decisions. This is not a strong assumption since it essentially entails that any change in our decision entails a change in the mean reward on average. This is why for MAB problems, this assumption holds for $\rho(\phi,\psi)=c\|\phi-\psi\|_1$ where $c>0$ is some constant that depends on the structure and prior (but independent of $\phi,\psi$).
>
> _Lack of interpretation / explanation_
> 1. This assumption means that there exists a small enough $\epsilon_0$-ball around every decision $p_1$ such that for any $p_2$ in this ball, the KL divergence is bounded by the norm $\rho$. This is essentially a regularity assumption over the decision space that prevents from the KL divergence to blow-up, which, in turn, can be obtained by requiring that $p_1$ and $p_2$ do not become arbitrarily small on their support. We have added further explanation on this assumption in the paper as well.
> 2. Theorem 3.4 is obtained by  substituting the bounds of Theorem 3.3 and the previously described selection process into Prop. 3.2. The last expression uses a general upper bound for $I(V;H_T)$ (which is the expression with $\mathrm{inf}_{\delta}$ ) and selects $\Phi$ to be a packing set of $\Pi$. The other two expressions use a different upper bound for $I(V;H_T)$, which utilizes the fact that we focus on a local packing set instead.
>
> _Minor Issues_
> 1. Thanks for the suggestion, we agree and changed the example to exclusively stick with an example of an online agent with access to a language model.
> 2. We have fixed the title.
> 3. We have fixed the reference to the Appendix.
>
> We hope that we have addressed your main concerns and if there are any other issues, we will gladly correct them.

---

> > ### Comment · Reviewer_smVV · 2024-11-19
> >
> > The authors have adequately addressed my concerns, and I will raise my score to a 6. I also realize that I misunderstood the rho assumption when I initially read the paper. I think it would be useful to add some explanation of that assumption to avoid confusion for future readers.

---

> > > ### Author Response · Authors · 2024-11-22
> > >
> > > Thank you for your response. We will add further explanation of the $\rho$ assumption to future revisions.

---

### Official Review · Reviewer_6gEU · 2024-11-05

**Soundness:** 3
**Presentation:** 3
**Contribution:** 3
**Rating:** 8
**Confidence:** 3

**Summary:**

The paper investigates the trade-off between information acquisition and regret minimization in sequential decision-making. It introduces a framework to quantify this trade-off, drawing on information-theoretic methods. The authors present novel Bayesian regret lower bounds that depend on the information accumulated by the agent, and they show how these quantify the relationship between external information and regret.
For brownie points, they show an application of their theory to question answering with LLMs.

**Strengths:**

- The paper presents an interesting information-theoretic approach to quantifying the regret-information trade-off
- The theoretical approach is rigorous, with clear definitions and proofs
- The paper is well-organized
- The paper holds high significance for fields involving sequential decision-making (online learning in particular)

**Weaknesses:**

- The assumption regarding information gathering being independent of task history could limit applicability in some environments

**Questions:**

- Can the authors comment on how the proposed regret bounds might extend to adversarial or non-Bayesian settings? Are there particular adjustments or challenges anticipated in these contexts?
- Could the authors comment on extensions to settings where the information depends on prior actions?

---

> ### Author Response · Authors · 2024-11-17
>
> We want to thank the reviewer for the detailed analysis and feedback. The following is a response addressing your concerns:
>
> (Q1) Since in non-Bayesian settings, there is no prior distribution, the mutual information $I(\pi^*;H_T)$ is not well defined, since it requires knowing the prior and posterior distributions. One possible way to extend this definition is to consider a constraint on the worst-case information gain at every step, $\min_{P}D_{KL}\left({P_{\pi^* \mid\pi,C}} ,{P_{\pi^*}}\right)$.
>
> (W1 + Q2) We agree that scenarios, where the collected information depends on previous actions, is an exciting future direction (as we mention in the conclusions section). One potential way to formulate this is by assuming that the contexts are sampled from a posterior distribution. That is, before an agent starts his/her interactions, the model $M \sim P$ is sampled and then the context $C_t$ is drawn from $C_t \sim P(\cdot \mid M,{\mathcal{H}}_t)$. The results presented in our paper can be relatively easily extended to this setting. However, this setting introduces different interesting questions. For example, should an agent in an online task explore a "bad" path to gain valuable external knowledge about a different one?

---

> > ### Comment · Reviewer_6gEU · 2024-11-26
> >
> > Thank you for answering my questions. I maintain my positive score.

---

### Official Review · Reviewer_oZWy · 2024-11-11

**Soundness:** 3
**Presentation:** 3
**Contribution:** 2
**Rating:** 6
**Confidence:** 3

**Summary:**

This paper studies regret minimization when extra information about the prior is revealed.
In particular, the authors consider contextual bandit problems, where at each round, the natural reveals some context and the algorithm needs to select actions based on the context. The authors consider a Bayesian set up, where there are some Bayesian prior on the context/reward. The external information reveals extra information about the prior. Under this formulation, the paper studies how external information affects learning and performance.

The paper proves both upper and lower bounds that depend on the amount of information an agent accumulates. The theoretical results demonstrate that information, measured in bits, can be directly traded off against regret, measured in reward. The paper also validates their findings through experiments with both traditional multi-armed bandits and a practical application involving large language models for question-answering tasks.

**Strengths:**

The problem formulated in this paper seems interesting, and it is interesting to see how information affects learning in general.
The paper also companies its theoretical results with experiments.

**Weaknesses:**

The technique is not the strong part of this paper.

**Questions:**

.

**Details Of Ethics Concerns:**

No concerns.

---

> ### Author Response · Authors · 2024-11-17
>
> We want to thank the reviewer for the detailed analysis and feedback. The following is a response addressing your concerns:
>
> (W1) The techniques in this paper include an adaptation of the method presented by [Yang and Barron, 1999] to an online setting. This includes utilizing covering and packing numbers to introduce a regret lower bound. We then introduce a modification of this method to handle problems with information constraint.

---

> > ### Comment · Reviewer_oZWy · 2024-11-26
> >
> > Thank you for your reponse. I want to keep my positive score.

---

### Meta-Review · Area_Chair_9F82 · 2024-12-20

**Metareview:**

This paper develops new information-theoretic methods for analyzing sequential decision-making algorithms. The methods are then used to recover existing lower bounds for a range of sequential decision-making problems. The reviewers gave overall very positive feedback and unanimously recommend acceptance of the paper. There are, however, some significant concerns about the presentation of the paper and I suggest that the authors implement the wide range of changes suggested by the reviewers.

**Additional Comments On Reviewer Discussion:**

There was some useful interaction between authors and reviewers and the reviewers adjusted their scores after the rebuttal and discussion period.

---

### Decision · Program_Chairs · 2025-01-22

Accept (Poster)